# Effectiveness of a Lifestyle Change Program on Insulin Resistance in Yaquis Indigenous Populations in Sonora, Mexico: PREVISY

**DOI:** 10.3390/nu15030597

**Published:** 2023-01-23

**Authors:** Alejandro Arturo Castro-Juarez, Araceli Serna-Gutiérrez, Heliodoro Alemán-Mateo, Ana Cristina Gallegos-Aguilar, Norma Alicia Dórame-López, Abraham Valenzuela-Sánchez, Diana Marcela Valenzuela-Guzmán, Rolando Giovanni Díaz-Zavala, Rene Urquidez-Romero, Julián Esparza-Romero

**Affiliations:** 1Diabetes Research Unit, Department of Public Nutrition and Health, Nutrition Coordination, Research Center for Food and Development (CIAD), A.C., Carretera Gustavo Enrique Astiazarán Rosas, No. 46 Col. La Victoria, Hermosillo 83304, Sonora, Mexico; 2Sociocultural Department, Technological Institute of Sonora, Cd. Obregon 85137, Sonora, Mexico; 3Department of Nutrition and Metabolism, Nutrition Coordination, Research Center for Food and Development (CIAD), A.C., Hermosillo 83304, Sonora, Mexico; 4Nutrition Health Promotion Center, Department of Chemical and Biological Sciences, University of Sonora, Hermosillo 83000, Sonora, Mexico; 5Department of Health Sciences, Institute of Biomedical Science, Autonomous University of Ciudad Juarez, Cd. Juarez 32310, Chihuahua, Mexico

**Keywords:** type 2 diabetes, prevention, insulin resistance, Yaqui indigenous

## Abstract

To evaluate the effectiveness of the healthy lifestyle promotion program for Yaquis (PREVISY) on insulin resistance in the short- and medium-term periods in adults who are overweight/obese and have an increased risk for diabetes. Using a translational research design, an intervention program was implemented in a sample of 93 Yaqui adult subjects. The effectiveness of PREVISY was evaluated by comparing the levels of Homeostasis Model Assessment of Insulin Resistance (HOMA-IR) and the Triglycerides-Glucose Index (TyG index) at 6 and 12 months using a paired *t*-test. Results: In the subjects who completed the program, a decrease in the HOMA-IR index (∆ = −0.91 and ∆ = −1.29, *p* ≤ 0.05) and the TyG index (∆ = −0.24 y ∆ = −0.20, *p* ≤ 0.05) was observed in the short- and medium-term period, respectively. Subjects with body weight loss ≥ 10% showed decreased levels of HOMA-IR (∆ = −3.32 and ∆ = −4.89, *p* ≤ 0.05) and the TyG index (∆ = −0.80 and ∆ = −0.60, *p* ≤ 0.05) at 6 and 12 months, respectively. A stronger benefit of the program was found in subjects with obesity (vs. overweight) and with high and very high risk of diabetes (vs. moderate risk) in IR markers (*p* ≤ 0.05). The PREVISY program demonstrated its effectiveness in the improvement of some markers of insulin resistance in Yaqui adults at risk of diabetes.

## 1. Introduction

Type 2 diabetes (T2D) is a chronic degenerative disease that has emerged as a worldwide public health problem, causing the death of 4.2 million people a year around the world [1]. According to the most recent data from the ENSANUT 2020 COVID-19, a prevalence of 15.7% was reported in Mexico, representing 12.8 million adults with T2D [2]. The biggest problem of T2D is the development of microvascular and macrovascular complications, causing high mortality [3]. Before COVID-19, T2D was positioned as the second cause of death in Mexican adults [4].

The pathophysiology of T2D is characterized by elevated blood glucose levels due to insulin resistance (IR) and insufficient insulin secretion by the pancreas [5]. Obesity, defined as excessive accumulation of fat or adipose tissue, is the main determinant of IR [6]. IR is a condition in which the cells of insulin-dependent tissues respond inadequately to insulin stimulation, due to alterations to the insulin receptor and to the effector molecules of the signaling cascade that block glucose transport into the cells. Currently, IR is considered an important risk factor for the development of T2D and cardiovascular diseases [7]. There are several methods to evaluate or determine IR [8]. The glucose clamp technique, with its two variants (hyperinsulinemic-euglycemic clamp and hyperglycemic clamp), is considered the gold standard for the measurement of IR [9,10]. However, for epidemiological studies, the glucose clamp technique is not a convenient method due to its invasive form, since it requires a highly specialized approach and time investment in its determination [10].

For epidemiological purposes, one of the methods most widely used is the Homeostasis Model Assessment of Insulin Resistance (HOMA-IR), which is highly correlated with the clamp technique and is calculated using fasting plasma glucose and serum insulin levels [11]. However, the cutoff of HOMA-IR to diagnose IR may vary between race and cannot be readily applied to all populations [12,13,14]. Another IR marker that has been used in recent years is the Triglycerides–Glucose Index (TyG index), which is considered a simple and low-cost method for epidemiological studies [15]. The TyG index has been associated with cardiovascular disease, T2D, hypertension, and metabolic syndrome [15].

Indigenous populations are characterized worldwide by having a low socioeconomic level, living in conditions of poverty and marginalization, and receiving poor or no health care [16]. In addition, in developed countries, indigenous groups experience a phenomenon of acculturation, which triggers the development of chronic diseases, such as T2D, showing high prevalence in some communities [17,18]. On the other hand, in developing countries such as Mexico, the acculturation of indigenous populations is a current process [19]. For this reason, some indigenous communities in Mexico, living a slightly Westernized lifestyle, present a low prevalence of T2D, while other ethnic groups with a more Westernized lifestyle present a high prevalence of this pathology [19,20,21,22]. A particular case is the Yaqui ethnic group, who reside in eight traditional towns, part of the municipalities of Guaymas, Bácum and Cajeme in the state of Sonora [23]. This indigenous group lived under a “traditional lifestyle”, which represented a protective factor against the development of obesity and chronic diseases. Their traditional diet was obtained in part from the collection of plants, roots and wild fruits for consumption, in addition to some cereals for the preparation of corn and wheat tortillas, legumes such as chickpeas and beans, and typical dishes such as “gallina pinta”, “wakabaki”, cheese broth, “caldillo de machaca”, among others [24,25,26]. They also had family gardens for self-consumption and for exchange among their neighbors [24]. Their diet ensured an adequate supply of nutrients, represented savings in the family economy and was of cultural interest [25]. Likewise, the main economic activities among its inhabitants were agriculture, fishing, livestock and the collection of firewood [27]. Both the food and the type of activities that they usually carried out in the past are different from the current ones. There are no data on the prevalence of obesity or chronic diseases when the Yaqui tribe lived the “traditional lifestyle” described. However, the same inhabitants of the community mention that these types of diseases did not exist or were quite rare at that time compared to what exists today.

Recent studies have reported changes in the lifestyle of the Yaqui community, with modifications in their diet, adopting hypercaloric foods in their diet and more sedentary activities with lower energy expenditure [27,28]. A recent study conducted in the eight traditional towns of the Yaqui community, using a probabilistic cross-sectional survey, reported a prevalence of overweight, obesity, and central obesity of 36.5%, 35.5%, and 76.0%, respectively. BMI was positively associated with higher modernity index, based on the possession of technological assets for the home and negatively associated with a higher consumption of a “prudent” dietary pattern consisting of traditional dishes, fruits, vegetables and low-fat dairy products and higher performing of vigorous intensity physical activity [29]. Previously, a different study reported a prevalence of T2D in the Yaqui community of 18.3% [20]. The public health crisis of the indigenous Yaquis is evident; therefore, the implementation of lifestyle changes that improve IR and the risk of developing T2D is of the utmost importance.

The healthy lifestyle promotion program for Yaquis (PREVISY) [30] is a program derived from the National Diabetes Prevention Program (NDPP) [31,32]. The PREVISY was implemented to evaluate the effectiveness of a health promotion program on obesity parameters and metabolic risk factors in the short- and medium-term periods in the Yaqui group. In this way, the effectiveness of the program was evaluated by comparing obesity parameters, physical activity and metabolic markers for the prevention of T2D in the Yaqui group [30]. However, it has not been evaluated whether the effectiveness of the PREVISY extends further by promoting changes in IR and its evaluation according to a greater loss of body weight, degree of obesity and greater risk of diabetes. Therefore, the aim of the study was to evaluate the effectiveness of the PREVISY on insulin resistance in the short- and medium-term in adults who are overweight/obese and have an increased risk for diabetes.

## 2. Materials and Methods

### 2.1. Study Design

In the present clinical study, a translational research design of a single cohort was analyzed with the PREVISY program [30]. The study recruited participants in July 2018, and the program was implemented from August 2018 to September 2019. The study protocol was approved by the Ethics Committee of the Research Center for Food and Development (CIAD), A.C. (CE/008-1/2018), registered on ClinicalTrials.com (NCT03599817), and all participants signed informed consent forms [30].

### 2.2. Subjects

The PREVISY included the participation of 93 Yaqui Indigenous adults from two locations (Loma de Guamúchil and Tórim) in the state of Sonora, Mexico. The inclusion criteria were men and women belonging to the Yaqui ethnic group, aged 20 to 65 years old, overweight/obese (BMI ≥ 25 kg/m^2^) [33] and at risk of diabetes (FINDRISC; score ≥ 12) [34]. On the other hand, subjects previously diagnosed with diabetes, uncontrolled hypertension (≥160/100 mmHg) and other self-reported diseases were excluded. Pregnant or lactating women, subjects with limitations in performing physical activities, and subjects who had participated in a similar program were also excluded. Subjects undergoing drug treatment for obesity or alterations in glucose tolerance and/or lipid profile were also excluded. The calculation of the sample size and the recruitment process have previously been reported [30].

### 2.3. Intervention Program

The implemented program (PREVISY) was adapted from the NDPP [31,32]. The PREVISY is a lifestyle change program that focuses on promoting healthy eating habits, physical activity and cognitive and behavioral changes in the participants. The goals of body weight loss of ≥5% and performance ≥150 min/week of physical activity of moderate–vigorous intensity were established [30]. The PREVISY contains an intensive phase (6 months) and a maintenance phase (6 months). In the first phase, sixteen weekly group sessions were given. From Sessions 1 to 4, topics related to healthy eating were taught, identifying foods with higher fat and caloric content and learning how to avoid consuming them, as well as knowing the food groups and recommended proportions. From Session 5 to 8, topics related to physical activity were taught, seeking a balance in the activities carried out and the food consumed. From Session 9 to 16, behavioral topics were taught that helped the participants to motivate themselves and not become discouraged, as well as strategies to maintain a healthy diet and perform physical activity. In the second phase, six monthly group sessions were given. The 6 sessions dealt with issues related to the first phase, with the aim of reinforcing what was learned and that the changes in diet, physical activity and behavior last in the medium-term. In total, the intervention program lasted 12 months. These sessions were taught by lifestyle coaches trained in the health area (bachelor’s and master’s in nutrition) [30]. The full participation of the program (completer subjects) was established with at least 80.0% of sessions received for each phase (≥13 sessions of the intensive phase and ≥5 sessions of the maintenance phase). The adaptation of the PREVISY to the Yaqui community, the content of the sessions, and the strategies to improve the adherence of the participants to the program have previously been reported in detail [30].

### 2.4. Measurement of Outcomes

To determine the effectiveness of PREVISY, measurements were taken before the implementation of the program (baseline measurements), at the end of the intensive phase, and at the end of the maintenance phase. Homa-IR and TyG index were evaluated by comparing the changes between the baseline and the 6-month measurement (short-term) and the changes between the baseline and the 12-month measurement (medium-term).

The measurements were recorded in the medical units of the Yaqui locations (Loma de Guamúchil and Tórim). The participants received instructions to attend the medical units in the morning following a 12 h fast for evaluation. A sample of 8 mL of peripheral venous blood was obtained, following previously reported methodology for extraction [35]. The serums were transported and analyzed in the laboratory of the Diabetes Research Unit of the Department of Public Nutrition and Health of the Research Center for Food and Development (CIAD), A.C.

### 2.5. HOMA-IR and TyG Index

We obtained biochemical components to determine the two IR indicators. First, fasting plasma glucose levels (expressed in mg/dL and mmol/L) were determined from the plasma sample extracted from peripheral venous blood using the Randox^®^ Glucose GOD-PAD technique, as previously reported [30]. Serum insulin levels (µU/mL) were obtained by enzyme immunoassay ELISA tests (DRG Diagnostics). The triglyceride values (mg/dL) were determined by the Randox^®^ Triglycerides GPO-PAP end-point enzymatic colorimetric method, as previously reported [30].

Using fasting plasma glucose and serum insulin values, the HOMA-IR values were estimated using the following formula [11]:(1)$$\mathrm{HOMA}-\mathrm{IR}=\frac{\mathrm{fasting}\;\mathrm{serum}\;\mathrm{insulin}\;\left(\frac{µU}{ml}\right)\;\times \;\mathrm{fasting}\;\mathrm{plasma}\;\mathrm{glucose}\;\left(\frac{mmol}{L}\right)}{22.5}$$

Similarly, using fasting plasma glucose and triglycerides values, the TyG index values were calculated using the following formula [36]:(2)$$TyG\;\mathrm{index}=ln\left[\mathrm{triglycerides}\;\left(\frac{mg}{dL}\right)\;\times \;\mathrm{fasting}\;\mathrm{plasma}\;\mathrm{glucose}\;\left(\frac{mg}{dL}\right)/2\right]$$

### 2.6. Anthropometry, Physical and Sociodemographic Measurements

Anthropometric, physical and sociodemographic measurements were collected to evaluate the baseline characteristics of the participants according to a previously published methodology [30]. Body weight (kg) was obtained using a portable digital scale with a maximum capacity of 200 kg ± 100 g (Seca 813). Height (m) was measured with a portable stadiometer (Seca 213) with an error range of 0.05 mm. Body mass index (BMI) was calculated by dividing a person’s body weight (kg) by the height in meters squared. Waist circumference was measured with anthropometric tape (GÜLICK, 0–150 cm scale). Systolic and diastolic blood pressure were determined using a digital baumanometer (Omron, model HEM-907XL) according to the established technique [37]. Overweight, obesity, and central obesity was diagnosed according to the criteria established by the World Health Organization (WHO) [33] and the International Diabetes Federation (IDF) [38], respectively. Sociodemographic and medical history questionnaires were also applied before starting the intervention program; the details of these questionnaires have been previously reported [30].

### 2.7. Statistical Analysis

Numerical variables are shown as the mean and standard deviation (mean ± SD). Categorical variables are expressed as percentages (%). The effectiveness of the program was evaluated by the change (∆) in the outcomes (baseline at 6 months and baseline at 12 months) using a paired *t*-test for variables with normal distribution or Wilcoxon rank-sum test for variables with non-normal distribution. An intention-to-treat analysis (ITT) and a protocol analysis (PA) were performed, the latter only with the subjects who completed the program. In the subjects who did not attend the evaluations at 6 and 12 months, the last value recorded was used [n = 4 (4.3%) at 6 months and n = 8 (8.6%) at 12 months]. Three sub-analyses, such as body weight loss, categories of overweight and obesity, and risk of diabetes (moderate risk vs. high and very high risk of diabetes), were carried out to evaluate the effect of the program on the values of HOMA-IR and the TyG index in the short- and medium-term. Additionally, a Pearson correlation was performed between the change in body weight and the changes in IR indicators in the completer subjects in the short- and medium-term. The statistical significance of all the analyses was set at a value of *p* ≤ 0.05. The analyses were carried out with Stata Statistical Software, version 16 (Copyright 1985–2019 Stata Corp LLP, College Station, TX, USA).

## 3. Results

Figure 1 shows the subject participation in the PREVISY, composed of 93 participants (75 women and 18 men). In total, 54 (58.0%) participants completed the intensive phase, while 52 (55.9%) participants completed the maintenance phase. It is important to mention the high response rate of the completer subjects and non-completer subjects of the program who attended the evaluations at 6 and 12 months, which was higher than 90.0% in both phases. This level of participation in both phases gives greater reliability to the results found in the statistical analyses.

Table 1 shows the baseline characteristics of the total participants of the PREVISY, stratified by completer subjects and non-completer subjects. It should be noted that some of these characteristics have already been published [30], showing serum insulin values, HOMA-IR and the TyG index as new results. A total of 80.6% of the participants in the program were women, with a mean age of 39.5 years. It is important to highlight the degree of risk of developing T2D, where 64.5% of the participants were classified as at high and very high risk of diabetes (according to the FINDRISC questionnaire), the average levels of fasting plasma glucose were 109.6 mg/dL, serum insulin was 23.8 µU/mL, triglycerides were 155.7 mg/dL, HOMA-IR was 6.6 and the TyG index was 8.9. There were no significant differences between completer subjects and non-completers subjects in any of the variables reported in Table 1, only in marital status (*p* ≤ 0.05).

An ITT analysis (all participants) was used to compare the changes in HOMA-IR levels and the TyG index at 6 and 12 months (Figure 2). Figure 2a shows that HOMA-IR decreased in both phases of the program (mean change = −0.25 ± 3.47 in the intensive phase and mean change = −0.54 ± 5.06 in the maintenance phase); however, these changes were not significant. On the other hand, Figure 2b shows that the TyG index decreased −0.14 ± 0.44 (*p* = 0.0003) at 6 months and −0.15 ± 0.50 (*p* = 0.0005) at 12 months. Figure 3 shows the same comparisons by PA (completer subjects). Figure 3a shows the changes in HOMA-IR values, with a reduction of −0.91 ± 3.08 (*p* = 0.03) at the end of the intensive phase and a reduction of −1.29 ± 4.00 (*p* = 0.02) at the end of the maintenance phase. Similarly, Figure 3b shows a decrease in the TyG index at 6 months (−0.24 ± 0.46, *p* = 0.0004) and at 12 months (−0.20 ± 0.52, *p* = 0.0008). In contrast, in non-completer subjects, the changes at the end of the intensive and maintenance phases from the baseline in HOMA-IR levels were +0.65 ± 3.79 (*p* = 0.28) and +0.41 ± 6.10 (*p* = 0.66), respectively. Regarding the TyG index, their observed changes from the baseline were −0.01 ± 0.39 (*p* = 0.87) and −0.08 ± 0.47 (*p* = 0.26), respectively (data not reported in tables).

To observe the effect of weight loss magnitude on the decrease in IR markers, a sub-analysis was performed by strata of loss of body weight, categorized by levels of compliance of weight loss goals (weight loss goal of 5 to 7%, of 7 to 10% and greater than or equal to 10%). Table 2 shows the results of HOMA-IR, and Table 3 shows the results of the TyG index.

Table 4 shows the sub-analysis to evaluate IR markers by categories of overweight and obese in the completer subjects. A greater benefit was shown in subjects classified with obesity, and a reduction in the HOMA-IR of −1.34 ± 3.62 (*p* = 0.03) was observed in the short term and of −2.06 ± 4.49 (*p* = 0.01) in the medium term. In addition, a decrease in the TyG index of −0.26 ± 0.53 (*p* = 0.005) at 6 months and of −0.29 ± 0.53 (*p* = 0.002) at 12 months was observed. Likewise, a similar sub-analysis was performed by categories of risk of diabetes in the completer subjects, according to the results of the FINDRISC questionnaire (two categories were performed: moderate risk and high and very high risk of diabetes) (Table 5). A greater benefit was shown in subjects classified as having a high and very high risk of diabetes, and a reduction in the HOMA-IR of −1.74 ± 3.18 (*p* = 0.003) in the short term and of −2.52 ± 3.75 (*p* = 0.0006) in the medium term was observed. In addition, a decrease in the TyG index of −0.32 ± 0.43 (*p* = 0.0001) at 6 months and of −0.25 ± 0.58 (*p* = 0.01) at 12 months was observed.

Additionally, the degree of correlation between the change in body weight and the change in IR markers was evaluated in the completer subjects of the PREVISY (Appendix A, Appendix A). As expected, positive correlations were observed in both phases of the study. In the intensive phase, body weight was correlated with HOMA-IR (r = 0.44, *p* = 0.0009), and with the TyG index (r = 0.49, *p* = 0.0002) (Appendix A, Appendix A). In the maintenance phase, body weight was correlated with HOMA-IR (r = 0.46, *p* = 0.005), and the TyG index (r = 0.55, *p* = 0.0000) (Appendix A, Appendix A).

Results regarding SBP and DBP in the completer subjects of the PREVISY indicated a slight increase in the SBP values of +3.18 ± 8.96 mmHg (*p* = 0.01) at the end of the intensive phase, but a reduction of −3.40 ± 8.73 mmHg (*p* = 0.007) at the end of the maintenance phase. Regarding DBP in the completer subjects, no change was observed at 6 months −0.07 ± 7.39 mmHg, (*p* = 0.94) while a decrease of −2.44 ± 7.50 mmHg (*p* = 0.02) was observed at 12 months, with respect to the basal level (data not reported in tables).

## 4. Discussion

The effectiveness of the PREVISY in the prevention of T2D has been evaluated by the improvement of obesity parameters and metabolic markers in the inhabitants of the Yaqui community in the state of Sonora, finding favorable results in the short and medium term [30]. In the present analysis, the effectiveness of the PREVISY on IR markers was evaluated by HOMA-IR and the TyG index. The cutoff of HOMA-IR to diagnose IR may vary from race to race. A study carried out in a Mexican population reported a cutoff point of 1.22 for the diagnosis of IR [14]. When comparing the baseline results of HOMA-IR in our participants, the average value was 6.6, indicating critical IR diagnosis. These results were expected since the selection of the participants was carried out carefully, seeking subjects that met the condition of having a high risk of T2D. It has been proposed that this type of program be applied as a priority specifically to patients with high risk of T2D [39,40].

The results of the effectiveness of the PREVISY on the HOMA-IR show a significant decrease of −0.91 and −1.29 at 6 and 12 months (*p* ≤ 0.05), respectively, in the completer subjects. Katula et al. (2011) implemented a lifestyle change program based on the Diabetes Prevention Program (DPP). The study included subjects with a BMI of 25–40 kg/m^2^ and prediabetes (fasting glucose: 95–125 mg/dL), including different ethnic/racial groups. The program consisted of weekly group sessions during the first 6 months and monthly sessions for the rest of the intervention; in total, the program lasted 12 months. The caloric intake goal of the participants was 1200–1800 kcal/day, and the physical activity goal was 180 min/week. In addition, the loss of body weight goal was 5 to 7%. Finally, the effectiveness of the program was evaluated, reporting a reduction in HOMA-IR of −1.71 (*p* ≤ 0.05) and −2.0 (*p* ≤ 0.05) at 6 and 12 months, respectively [41], which were greater than those in our study. Similarly, Herder et al. (2009), using a subsample of 265 subjects of the intervention group of the Finnish Diabetes Prevention Study (Finnish DPS), evaluated the program at 12 months. The study included subjects with BMI ≥25 kg/m^2^ and glucose intolerance (two tests were performed: a glucose tolerance test at 2 h (140–200 mg/dL) and fasting glucose (<140 mg/dL)). The program consisted of bimonthly sessions with a nutritionist, and the participants had to perform ≥30 min/day of physical activity, reduce fat intake to <30%, fiber consumption ≥15 g/1000 kcal and lose ≥5% body weight. At the end of the intervention program (12 months), a decrease of −0.80 in the values of HOMA-IR was reported (*p* ≤ 0.05) [42], which was similar to the results obtained in our study.

TyG index as a marker of IR can also vary according to the population studied; in the Mexican population, a cutoff point greater than or equal to 8.17 has been reported as a diagnosis of IR [14]. The baseline levels of index TyG in our study were an average of 8.9, which confirms what was found with HOMA-IR values, explained above. Furthermore, it has been reported that high values of the TyG index are associated with an increased risk of T2D [43,44]. The results of the effectiveness of the PREVISY on the TyG index showed a significant decrease of −0.24 and −0.20 at 6 and 12 months (*p* ≤ 0.05), respectively. Currently, there are no studies that have evaluated the effectiveness of lifestyle change programs on TyG index values. However, it has been reported that changes in this indicator may be more decisive in predicting the development of T2D than an increase in body weight [45]. A meta-analysis carried out in a systematic review of cohort studies showed that subjects classified in the upper quartile of the TyG index had up to 2.44 (HR: 2.44, 95% CI: 2.17–2.76) times more risk of developing T2D than subjects classified in the lower quartile [46]. In this sense, the improvements achieved in the PREVISY on the TyG index can translate into a significant decrease in the risk of T2D.

Programs that promote healthy lifestyles have a greater benefit in subjects at higher risk [39]. Our study showed a greater benefit in subjects classified as obese (vs. overweight), with significant decreases of up to 28.7% in HOMA-IR and 3.2% in the TyG index. Additionally, a greater benefit was observed in subjects classified as at high and very high risk of diabetes (vs. moderate risk), with significant decreases of up to 35.3% in HOMA-IR and 3.5% in the TyG index.

Stronger results were observed in IR markers when subjects achieved the goal of body weight loss, with 14.6% to 54.4% decreases in HOMA-IR and 1.7% to 8.8% in the TyG index. However, some improvements were not significant due to the small sample size in the strata. The required sample size was calculated in the stratum of loss of body weight of 7 to 10%, which was the closest to reaching statistical significance. In the case of HOMA-IR, the necessary sample was ≥9 subjects, and in the case of the TyG index, the necessary sample was ≥16 subjects. Nevertheless, the improvements in IR markers were relevant to the metabolic health of the participants. The observed decrease in HOMA-IR and the TyG index may be reflected in the improvement of fasting glucose with reductions of −10.7 mg/dL and −14.4 mg/dL to short- and medium-term PREVISY [30]. Likewise, it is important to mention that there were decreases in serum insulin levels during the short- (−0.59 µU/mL, *p* > 0.05) and medium-term (−1.50 µU/mL, *p* > 0.05) in the completer subjects (data not reported in tables), contributing to a metabolic improvement in the hyperinsulinemic state of the participants.

Regarding the results obtained from blood pressure, the PREVISY was not aimed at directly reducing these parameters (SBP and DBP), since it was not a selection criterion during recruitment and, consequently, as is clear from Table 1, most of the participants had normal blood pressure values (81.0%) based on The Eighth Joint National Committee (JNC-8) [47]. Despite this, favorable results were observed in SBP and DBP, especially in the medium-term, confirming that this type of program may also contribute to the improvement of these parameters. However, it is expected that there may be a greater benefit in subjects with elevated blood pressure levels compared to subject with normal values.

It is also important to mention the effectiveness of the PREVISY on other evaluated parameters in the completer subjects. A significant reduction in body weight of −3.9 ± 4.5 and of −3.9 ± 6.2 kg was observed at the end of the intensive and maintenance phase, respectively. A significant decrease in waist circumference (−4.2 ± 5.8 and 3.6 ± 6.1 cm) and BMI (−1.5 ± 1.8 and −1.5 ± 2 kg/m^2^) was also observed at 6 and 12 months, respectively. Regarding lipid profile, a significant improvement was observed at the end of the intensive phase in total cholesterol (−12.6 ± 22.2 mg/dL), LDL-c (−13.6 ± 19.6 mg/dL), triglycerides (−22.7 ± 63.2 mg/dL) and HDL-c (+6.0 ± 9.8 mg/dL). Likewise, improvements, although not significant, were observed at the end of the maintenance phase in some of these parameters, with −1.8 ± 22.9 mg/dL in total cholesterol, −11.8 ± 62.0 mg/dL in triglycerides and +2.4 ± 8.0 mg/dL in HDL-c [30].

The importance of reducing body weight in this type of program is very relevant. PREVISY had an average reduction in body weight of −3.9 kg (4.5%) in the short- and medium-term in the completer subjects [30]. Correlation analysis indicated positive correlations, to a moderate degree, between the change in body weight and the change in HOMA-IR, and the change in the TyG index in the short- and medium-term in the completer subjects. This shows that a greater change in body weight contributed to a greater change in IR markers. It has been reported that a 3% decrease in body weight can lead to improvements in IR markers, and thus to a decrease in the incidence of T2D [48]. IR is a direct link between obesity and T2D, and its decrease contributes to the improvement of glucose levels, lipid profiles and markers of vascular inflammation [42,49,50]. Some programs that promote a healthy lifestyle have shown efficacy in reducing body weight and, therefore, produce a lower incidence of T2D [51,52]. In general, IR markers have a greater variation from person to person than other anthropometric and metabolic markers [53], so their evaluation is relevant for the prevention of T2D.

The PREVISY focused on promoting changes in lifestyle habits, mainly those related to food and physical activity. In this way, the participants learned how to moderate their fat intake, substituting less healthy foods for healthier ones; they learned about the portions they should consume, how to count the calories consumed and about energy balance. Regarding physical activity, the participants learned about the importance of physical activity, as well as the concepts of type, intensity and time of physical activity for better health. IR is caused by metabolic changes associated with obesity and with the development of T2D [54]. Some metabolic changes generate hyperphosphorylation of serine and threonine residues of IRS proteins (insulin receptor substrate), reducing tyrosine phosphorylation and thus reducing the interaction with PI3K (phosphatidylinositol 3-kinase) and altering the phosphorylation and activation of AKT kinase (serine/threonine-protein). The inhibition of this signaling cascade impedes the glucose transport to the interior of the cells of insulin-dependent tissues (such as adipose, muscle and liver tissue), contributing to IR and subsequently to the development of T2D and cardiovascular diseases [55]. The beneficial effects found in PREVISY reflected in the levels of HOMA-IR and in the TyG index may demonstrate improved function in the cell signaling cascade, thus reducing IR and the risk of T2D.

Some strengths of the study were the translation of a lifestyle change program to an indigenous community characterized by low resources and a high risk of diabetes, which supports the effectiveness of this type of program in the “real world”. Our study supports the use of the TyG index as a method to evaluate IR, being the first intervention study to evaluate this parameter in the short- and medium-term in an indigenous population. It is also worth mentioning that the recruitment phase captured the subjects with the highest risk of T2D. In addition, the high response rate of the participants during the different evaluation times (6 and 12 months) gave greater strength to the results presented by ITT and PA. The low percentage of retention could be a limitation of the study; however, other programs have reported similar and higher rates [39,56,57], so there is room for improvement.

## 5. Conclusions

The PREVISY is effective in the improvement of IR markers, contributing to a direct appreciation of the benefit of this type of intervention in the reduction of the risk of T2D. In turn, we observed that a greater loss of body weight promotes a greater change in IR. In the same way, the subjects classified as obese (vs. overweight) and the subjects with a high and very high risk of diabetes (vs. moderate risk) had a greater benefit from the program, along with a greater change in IR markers. The results support the use of programs focused on promoting healthy eating habits and physical activity as the gold standard for the prevention of chronic diseases. At the same time, they support the translation of this type of program to vulnerable populations such as the Yaqui tribe. Its implementation to a greater number of subjects of this indigenous community is reliable. Therefore, a healthy lifestyle change program should be the first option for the prevention of chronic diseases such as T2D. It is important to highlight that this type of intervention could be part of public policy in the health sector through the integration of a multidisciplinary team, which must include expert nutritionists and physical activity educators as a fundamental part of the team.

## Figures and Tables

**Figure 1 nutrients-15-00597-f001:**
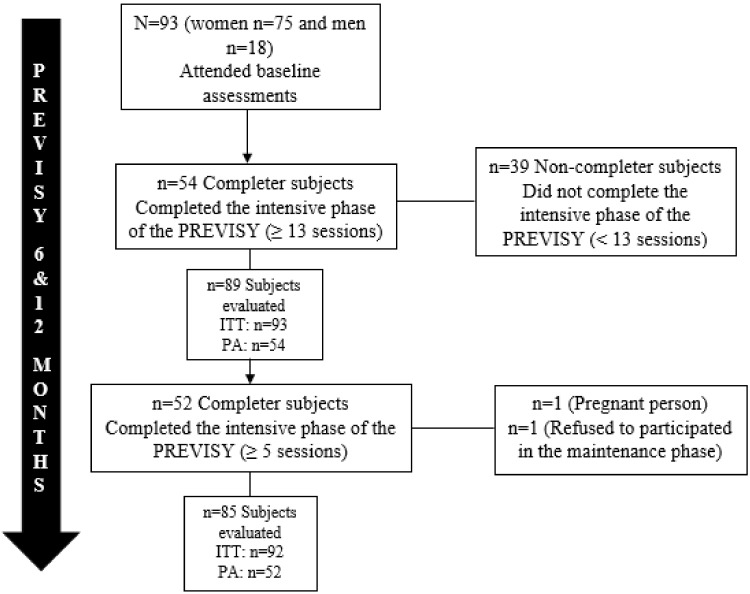
Participation of the subjects in the intensive and maintenance phase of the PREVISY. ITT: intention-to-treat analysis; PA: protocol analysis.

**Figure 2 nutrients-15-00597-f002:**
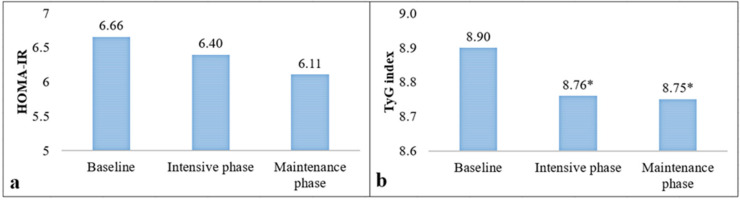
Comparison of HOMA-IR (**a**) and TyG index (**b**) levels in the intensive and maintenance phase of the PREVISY by intention-to-treat analysis (ITT). HOMA-IR: Homeostasis Model Assessment of Insulin Resistance; TyG index: Triglycerides-Glucose Index. Expressed as the mean; *: Significant differences with respect to the baseline measurement (*p* ≤ 0.05). Paired *t*-test.

**Figure 3 nutrients-15-00597-f003:**
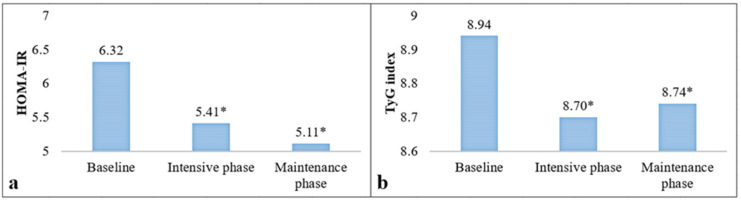
Comparison of HOMA-IR (**a**) and TyG index (**b**) levels in the intensive and maintenance phase of the PREVISY using protocol analysis (PA). HOMA-IR: Homeostasis Model Assessment of Insulin Resistance; TyG index: Triglycerides–Glucose Index. Expressed as the mean; *: Significant differences with respect to the baseline measurement (*p* ≤ 0.05). Paired *t*-test.

**Table 1 nutrients-15-00597-t001:** Baseline characteristics of the participants.

Characteristics	Total	Completers	Non-Completers	*p*
N	93	54	39	
Gender ‡ Women Men				1.00
	75 (80.6)	44 (81.5)	31 (79.5)	
	18 (19.4)	10 (18.5)	8 (20.5)	
Age (years) †	39.5 ± 11.2	40.1 ± 10.5	38.6 ± 12.1	0.54
Scholarship ‡ Primary Secondary school High school University				0.59
	27 (29.1)	13 (24.1)	14 (35.9)	
	36 (38.7)	21 (38.9)	15 (38.5)	
	15 (16.1)	10 (18.5)	5 (12.8)	
	15 (16.1)	10 (18.5)	5 (12.8)	
Civil status ‡ Single, widower or separate Married or free union				0.008
	31 (33.3)	24 (44.4)	7 (18.0)	
	62 (66.7)	30 (55.6)	32 (82.0)	
Overweight ‡	28 (30.1)	19 (35.2)	9 (23.1)	0.25
Obesity ‡	65 (69.9)	35 (64.8)	30 (76.9)	0.25
Central obesity ‡	92 (98.9)	53 (98.2)	39 (100.0)	1.00
Previous diagnosis of HT ‡	13 (14.0)	8 (14.8)	5 (12.8)	1.00
Risk of T2D (FINDRISC) ‡ Moderate risk High and very high risk				0.51
	33 (35.5)	21 (38.9)	12 (30.8)	
	60 (64.5)	33 (61.1)	27 (69.2)	
Body weight (kg) †	85.9 ± 14.6	85.0 ± 13.5	87.1 ± 16.1	0.50
WC (cm) †	104.4 ± 10.6	103.3 ± 10.1	106.0 ± 11.3	0.23
BMI (kg/m^2^) †	33.2 ± 5.2	32.6 ± 4.6	34.1 ± 6.0	0.17
SBP (mmHg) †	115.7 ± 13.0	116.6 ± 11.0	114.5 ± 15.5	0.45
DBP (mmHg) †	73.3 ± 9.4	73.6 ± 8.6	72.8 ± 10.5	0.67
Fasting glucose (mg/dL) †	109.6 ± 26.2	110 ± 28.9	108.7 ± 22.4	0.77
Triglycerides (mg/dL) †	155.7 ± 87.7	159.1 ± 75.2	151.1 ± 103.4	0.66
Serum insulin (µU/mL) †	23.8 ± 11.7	22.4 ± 10.9	25.8 ± 12.6	0.16
HOMA-IR †	6.6 ± 4.2	6.3 ± 3.9	7.1 ± 4.6	0.37
TyG index †	8.9 ± 0.5	8.9 ± 0.5	8.8 ± 0.5	0.43

FINDRISC: moderate risk (score 12–14), high risk (score 15–20) and very high risk (score > 20). Note: for the analysis, the categories of high risk and very high risk were combined. T2D: Type 2 diabetes; HT: Hypertension; WC: Waist circumference; SBP: Systolic blood pressure; DBP: Diastolic blood pressure; HOMA-IR: Homeostasis Model Assessment of Insulin Resistance; TyG index: Triglycerides–Glucose Index. ‡: Expressed as the n (%); †: Expressed as the mean ± SD; *p* value: T-test for independent samples (normally distributed variables); X^2^ test and Fisher’s exact tests (categorical variables).

**Table 2 nutrients-15-00597-t002:** Comparison of HOMA-IR levels by body weight loss stratum.

		HOMA-IR Levels								
Body weight goal (%)	n	Baseline	Intensive phase	∆	*p* value	n	Baseline	Maintenance phase	∆	*p* value
5 to 7	11	5.54 ± 3.45	4.73 ± 2.58	−0.81 ± 2.01	0.21	11	8.51 ± 7.76	5.91 ± 4.22	−2.60 ± 5.30	0.13
7 to 10	6	9.76 ± 4.98	5.53 ± 2.46	−4.23 ± 4.39	0.06	4	8.92 ± 6.02	4.15 ± 1.83	−4.77 ± 4.24	0.10
≥10	7	8.27 ± 2.79	4.94 ± 2.02	−3.32 ± 1.82	0.002	10	8.98 ± 3.31	4.09 ± 1.59	−4.89 ± 3.93	0.003

Expressed as the mean ± SD; ∆: Change mean ± SD; *p* value: Paired *t*-test.

**Table 3 nutrients-15-00597-t003:** Comparison of TyG index levels by body weight loss stratum.

		TyG Index Levels								
Body weight goal (%)	n	Baseline	Intensive phase	∆	*p* value	n	Baseline	Maintenance phase	∆	*p* value
5 to 7	11	9.13 ± 0.75	8.87 ± 0.59	−0.26 ± 0.32	0.02	11	9.17 ± 0.57	9.00 ± 0.58	−0.16 ± 0.39	0.20
7 to 10	6	9.15 ± 0.63	8.81 ± 0.37	−0.33 ± 0.46	0.14	4	9.02 ± 0.54	8.47 ± 0.28	−0.55 ± 0.47	0.10
≥10	7	9.14 ± 0.41	8.34 ± 0.38	−0.80 ± 0.35	0.001	10	8.90 ± 0.59	8.30 ± 0.31	−0.60 ± 0.65	0.01

Expressed as the mean ± SD; ∆: Change mean ± SD; *p* value: Paired *t*-test.

**Table 4 nutrients-15-00597-t004:** Comparison of IR markers in the completer subjects with overweight or obesity.

Outcomes	Overweight (n = 19)	Obesity (n = 35)	∆ Overweight	∆ Obesity
HOMA-IR Baseline Intensive phase Maintenance phase				
	4.77 ± 2.43	7.17 ± 4.37		
	4.63 ± 2.28	5.83 ± 2.88	−0.14 ± 1.50	−1.34 ± 3.62 *
	4.87 ± 2.32	5.24 ± 3.21	+0.16 ± 2.35	−2.06 ± 4.49 *
TyG index				
Baseline Intensive phase Maintenance phase	8.99 ± 0.67	8.91 ± 0.52		
	8.81 ± 0.58	8.65 ± 0.48	−0.18 ± 0.30 *	−0.26 ± 0.53 *
	8.96 ± 0.60	8.62 ± 0.47	−0.02 ± 0.47	−0.29 ± 0.53 *

HOMA-IR: Homeostasis Model Assessment of Insulin Resistance; TyG index: Triglycerides–Glucose Index. Expressed as the mean ± SD; ∆: Change mean ± SD; *: *p* ≤ 0.05. Paired *t*-test.

**Table 5 nutrients-15-00597-t005:** Comparison of IR markers in the completer subjects with moderate risk or high and very high risk of diabetes by the FINDRISC questionnaire.

Outcomes	Moderate Risk (n = 21)	High and Very High Risk (n = 33)	∆ Moderate Risk	∆ High and Very High Risk
HOMA-IR				
Baseline Intensive phase Maintenance phase	5.08 ± 2.78	7.12 ± 4.39		
	5.46 ± 2.90	5.37 ± 2.65	+0.38 ± 2.48	−1.74 ± 3.18 *
	5.71 ± 3.50	4.74 ± 2.47	+0.68 ± 3.65	−2.52 ± 3.75 *
TyG index				
Baseline Intensive phase Maintenance phase	8.94 ± 0.70	8.94 ± 0.49		
	8.84 ± 0.62	8.61 ± 0.43	−0.09 ± 0.48	−0.32 ± 0.43 *
	8.82 ± 0.68	8.69 ± 0.43	−0.11 ± 0.42	−0.25 ± 0.58 *

FINDRISC: moderate risk (score 12–14), high risk (score 15–20) and very high risk (score > 20). Note: for the analysis, the categories of high risk and very high risk were combined. HOMA-IR: Homeostasis Model Assessment of Insulin Resistance; TyG index: Triglycerides–Glucose Index. Expressed as the mean ± SD; ∆: Change mean ± SD; *: *p* ≤ 0.05. Paired *t*-test.

## Data Availability

The data obtained and analyzed in the study can be requested, depending on their use, from the corresponding author.

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
