# Peer review of "Effectiveness of a Lifestyle Change Program on Insulin Resistance in Yaquis Indigenous Populations in Sonora, Mexico: PREVISY"

_nutrients, 2023, doi:10.3390/nu15030597_

Round 1
Reviewer 1 Report
Thank you for submitting the manuscript "Effectiveness of a lifestyle change program on insulin resistance in Yaquis indigenous populations in Sonora, Mexico: PREVISY" to Nutrients.
The researchers evaluated the effectiveness of PREVISY as a T2D prevention protocol. Overall, the manuscript is well written, the research is relevant, and the sampling appears to have been loaded with scientific requirements in mind. I have small considerations to improve the quality of the manuscript.
Introduction:
as the abstract reports results related to obesity and overweight, it would be interesting to reorganize the introduction item bringing the description of obesity and overweight as well as its data at global levels and specifically in the population's ethnicity in the first place. This issue is so important in the case of this study since individuals with diabetes were even excluded from the research.
Or even characterize the indigenous population in the first place and link the anthropometric change of this population with the arrival of civilization.
Material and methods
It is interesting to include which sample was recruited from the population, that is, from how many people were the 93 Yaqui indigenous adults recruited? This will give the reader an idea of the size of the evaluated sample related to the size of the population.
Line#229: improve the readability of this line, it looks erased.
Line#235: Correct the citation of the reference.
Author Response
Please see the attachment
Response to Reviewer 1 Comments
Thank you for submitting the manuscript "Effectiveness of a lifestyle change program on
insulin resistance in Yaquis indigenous populations in Sonora, Mexico: PREVISY" to
Nutrients.
The researchers evaluated the effectiveness of PREVISY as a T2D prevention protocol.
Overall, the manuscript is well written, the research is relevant, and the sampling appears to
have been loaded with scientific requirements in mind. I have small considerations to
improve the quality of the manuscript.
Point 1:
Introduction:
as the abstract reports results related to obesity and overweight, it would be interesting to
reorganize the introduction item bringing the description of obesity and overweight as well
as its data at global levels and specifically in the population's ethnicity in the first place. This
issue is so important in the case of this study since individuals with diabetes were even
excluded from the research.
Or even characterize the indigenous population in the first place and link the anthropometric
change of this population with the arrival of civilization.
Response 1:
The central theme of the manuscript is the prevention of type 2 diabetes (T2D) where insulin
resistance and obesity play a major role as triggers for its development. That is why the
introduction section focuses on the problem T2D, but also handles relevant information on
overweight and obesity and its determinants in the Yaqui ethnic group in the lines #92-100,
as well as the prevalence of T2D in the community. The introduction section begins by
providing an overview in national context of T2D, which is relatively important, and then
highlights the main trigger that leads to the development of the disease. Then comes the core
part where the study group is characterized, providing relevant, specific and recent
information on said interest group (Yaqui indigenous). We believe that the introduction
section handles information that goes from the general to the most specific, leading the reader
to the objective of the investigation.
Point 2:
Material and methods
It is interesting to include which sample was recruited from the population, that is, from how
many people were the 93 Yaqui indigenous adults recruited? This will give the reader an idea
of the size of the evaluated sample related to the size of the population.
Response 2:
Thanks for the observation, for the design and objective of our study, the sample evaluated
did not seek to be representative of the adults of the Yaqui communities. This is a clinical
intervention study aimed at evaluating the effectiveness of the program in improving insulin
resistance markers in participants at risk of T2D, for which 175 subjects were screened during
recruitment to identify and complete the estimated sample in these types of participants. In
this way, it was sought that the program would benefit subjects who, if not attended to, could
develop T2D or other associated diseases in the short or medium term. Looking at the results
obtained, it is clear that the benefits of PREVISY were much better on markers of insulin
resistance in subjects at higher risk of diabetes.
Point 3:
Line#229: improve the readability of this line, it looks erased.
Response 3:
Line #229 is part of the sentence that include from lines 228-229, that same sentence is the
idea that concludes the previous sentence that include from lines 225-228. We believe that
there is a good wording in the set of ideas that are handled in the paragraph. Similarly, line
229 ends with the word "ITT analysis" (intention-to-treat analysis), which is abbreviated, but
defined previously in the manuscript.
However, based on your observation, we deleted in line 229 the following phrase: mainly
the ITT analysis.
Point 4:
Line#235: Correct the citation of the reference.
Response 4:
Corrected the citation on line 235, appearing like the other citations and as required by the
journal format ...already been published [30],...

Reviewer 2 Report
The study focused on the evaluation of the effectiveness of the healthy lifestyle promotion program on insulin resistance among overweight/obese adults form the Yaqui ethnic group in Mexico. The article covers very important aspects of health problems in indigenous population.
Please address the following issues:
– The authors cite 5 studies of their own. They also point out that some results have already been published (lines 234-235). However, Table 1 contains almost the same data as published in Table 2 in the publication cited number 30. Please revise the results presented in this study to avoid the duplication.
– Lines 278-290, 301-312 please improve the presentation of the results so that they are not presented two times, in descriptive form and in a table.
– The paper presents results indicating the effectiveness of the PREVISY program. However, there is no explanation which factors influenced the final result - a change in diet quality, energy and macronutrients intakes, macronutrients composition of the diet, higher level of physical activity, or perhaps other factors. Please explain the potential mechanisms of these changes.
– Please provide information on the lifestyle changes that can explain the effectiveness of the program – reduction in body mass, HOMA IR, TyG and blood pressure.
– Please explain how to implement the results of the study in practice. What should be done to prevent non-communicable diseases among the Yaqui ethnic group.
Author Response
Please see the attachment
Response to Reviewer 2 Comments
The study focused on the evaluation of the effectiveness of the healthy lifestyle promotion
program on insulin resistance among overweight/obese adults form the Yaqui ethnic group
in Mexico. The article covers very important aspects of health problems in indigenous
population.
Please address the following issues:
Point 1:
The authors cite 5 studies of their own. They also point out that some results have already
been published (lines 234-235). However, Table 1 contains almost the same data as published
in Table 2 in the publication cited number 30. Please revise the results presented in this study
to avoid the duplication.
Response 1:
The work team for the investigation is a specialist in the area of diabetes research focused on
indigenous communities for several decades, some data that is cited is due to the seniority
that has been working in different ethnic groups of Sonora, Mexico. The work team has
recently studied the Yaqui indigenous community, so most of the most recent and relevant
information has been published by the work team itself.
In lines 234-235 it is said that certain information has already been published, however, it is
also mentioned that there are also new data such as HOMA-IR, serum insulin and TyG index
that complement said information. Table 1 (Basal characteristics of the participants) where
this information is shown helps us to better characterize the study group, they are only
baseline characteristics that are relevant to understanding the lifestyle and risk factors of the
community. These are not results that demonstrate the effectiveness of the program, they are
only to characterize the population. As for the response variables that show the effectiveness
of the program, they are results that have never been published before, in addition to the sub-
analyses that are very important.
Point 2:
Lines 278-290, 301-312 please improve the presentation of the results so that they are not
presented two times, in descriptive form and in a table.
Response 2:
For the results described in lines 278-290, it was decided to delimit the explanation given
since the information contained in tables 2 and 3 was duplicated, leaving now only this one:
“To observe the effect of weight loss magnitude on the decrease in IR markers, a sub-analysis
was performed by strata of loss of body weight, categorized by levels of compliance of
weight loss goals (weight loss goal of 5 to 7%, of 7 to 10% and greater than or equal to 10%).
Table 2 shows the results of HOMA-IR and the table 3 shows the results of TyG index”. In
this way the reader will be able to see the information in these tables.
Regarding lines 301-312, we decided to leave the information as it is written since they do
not duplicate the information contained in tables 4 and 5. He cites some results that are most
relevant for a better understanding of the reader.
Point 3:
The paper presents results indicating the effectiveness of the PREVISY program. However,
there is no explanation which factors influenced the final result - a change in diet quality,
energy and macronutrients intakes, macronutrients composition of the diet, higher level of
physical activity, or perhaps other factors. Please explain the potential mechanisms of these
changes.
Response 3:
To emphasize the reason for the favorable results, information was integrated about the
lifestyle changes that the participants adopted and that are important for the improvement of
the outcomes. The information was integrated into the lines 438-444.
“The PREVISY focused on promoting changes in lifestyle habits, mainly those related to
food and physical activity. In this way, the participants learned how to moderate their fat
intake, substituting less healthy foods for healthier ones, they learned about the portions they
should consume, how to count the calories consumed and about energy balance. Regarding
physical activity, the participants learned about the importance of physical activity, as well
as the concepts of type, intensity and time of physical activity for better health”.
Point 4:
Please provide information on the lifestyle changes that can explain the effectiveness of the
program – reduction in body mass, HOMA IR, TyG and blood pressure.
Response 4:
We believe that based on the integrated information requested in the previous point, the point
proposed by the reviewer can be answered. The information was integrated into the lines
438-444.
“The PREVISY focused on promoting changes in lifestyle habits, mainly those related to
food and physical activity. In this way, the participants learned how to moderate their fat
intake, substituting less healthy foods for healthier ones, they learned about the portions they
should consume, how to count the calories consumed and about energy balance. Regarding
physical activity, the participants learned about the importance of physical activity, as well
as the concepts of type, intensity and time of physical activity for better health”.
Point 5:
Please explain how to implement the results of the study in practice. What should be done to
prevent non-communicable diseases among the Yaqui ethnic group.
Response 5:
The conclusions section briefly addresses the information requested in lines 473-477:
“The results support the use of programs focused on promoting healthy eating habits and
physical activity as the gold standard for the prevention of chronic diseases. At the same
time, it supports the translation of this type of program to vulnerable populations such as the
Yaqui tribe. Therefore, its implementation to a greater number of subjects of this indigenous
community is reliable”.
We complement said information with the following (lines 477-482):
“Therefore, a healthy lifestyle change program should be the first option for the prevention
of chronic diseases such as T2D. It is important to highlight that this type of interventions
could be part of a public policy in the health sector through the integration of a
multidisciplinary team, which must include expert nutritionists and physical activity
educators as a fundamental part of the team”.
